# Sequence Analysis and Functional Verification of the Effects of Three Key Structural Genes, *PdTHC2’GT*, *PdCHS* and *PdCHI*, on the Isosalipurposide Synthesis Pathway in *Paeonia delavayi* var. *lutea*

**DOI:** 10.3390/ijms23105696

**Published:** 2022-05-19

**Authors:** Hongzhu Zou, Lulu Han, Meng Yuan, Mengjie Zhang, Lin Zhou, Yan Wang

**Affiliations:** Key Laboratory of Tree Breeding and Cultivation of National Forestry and Grassland Administration, Research Institute of Forestry, Chinese Academy of Forestry, Beijing 100091, China; zouhongzhu1123@163.com (H.Z.); hanlulu0727@163.com (L.H.); yuanmeng00829@163.com (M.Y.); 15032327681@163.com (M.Z.)

**Keywords:** *Paeonia delavayi* var. *lutea*, isosalipurposide, *PdTHC2’GT*, *PdCHS*, *PdCHI*, functional verification

## Abstract

Isosalipurposide (ISP) is the most important yellow pigment in tree peony. In ISP biosynthesis, *CHS* catalyzes 1-molecule coumaroyl-CoA and 3-molecule malonyl-CoA to form 2′,4′,6′,4-tetrahyroxychalcone (THC), and THC generates a stable ISP in the vacuole under the action of chalcone2′-glucosyltransferases (*THC2′GT*). In tree peony, the details of the *THC2’GT* gene have not yet been reported. In this study, the candidate *THC2’GT* gene (*PdTHC2’GT*) in *Paeonia delavayi* var. lutea was screened. At the same time, we selected the upstream *CHS* gene (*PdCHS*) and the competitive *CHI* gene (*PdCHI*) to study the biosynthesis pathway of ISP. We successfully cloned three genes and sequenced them; subcellular localization showed that the three genes were located in the nucleus and cytoplasm. The overexpression of *PdTHC2’GT* in tobacco caused the accumulation of ISP in tobacco petals, which indicated that *PdTHC2’GT* was the key structural gene in the synthesis of ISP. After the overexpression of *PdCHS* and *PdCHI* in tobacco, the accumulation of anthocyanins in tobacco petals increased to different degrees, showing the role of *PdCHS* and *PdCHI* in anthocyanin accumulation. The analysis of *NtCHS* and *NtCHI* of transgenic tobacco lines by qRT-PCR showed that the *THC2’GT* gene could increase the expression of *CHS*. *THC2’GT* and *CHI* were found to be competitive; hence, the overexpression of *THC2’GT* could lead to a decrease in *CHI* expression. The *CHS* gene and *CHI* gene could increase the expression of each other. In conclusion, we verified the key structural gene *PdTHC2’GT* and studied the operation of the genes in its upstream and competitive pathway, providing a new perspective for the biosynthesis of ISP and new candidate genes for the directional breeding of tree peony.

## 1. Introduction

Tree peony (*Paeonia suffruticosa* Andrew) is a traditional Chinese flower that is valued by people all over the world because of its beautiful flowers. Tree peonies with yellow flowers have always been valuable and are also a current breeding direction. *P. delavayi* is a special species due to its various petal colors, including yellow, orange, red, purple-red, and dark red [1]. *P. delavayi* var. *lutea* is its yellow variation. The more successful yellow tree peony varieties on the market, such as the ‘High Noon’, ‘Alice Harding’, and Itoh hybrid series, have a *P. delavayi* var. *lutea* pedigree. Stable pure yellow petals have made *P. delavayi* var. *lutea* a suitable material for researching the formation of yellow flowers in tree peony. *P. delavayi* and *P. suffruticosa* belong to different subgroups, which makes hybridization difficult and results in a long breeding cycle. Hybrids also easily inherit the characteristics of small flowers and drooping flower heads from *P. delavayi*. Therefore, molecular breeding is the best way to obtain yellow-flowered tree peony. However, the molecular mechanism of yellow color synthesis in *P. delavayi* petals is not clear. Analyzing the molecular regulatory mechanism of yellow color synthesis in *P. delavayi* petals will be helpful for identifying the key genes involved in yellow anthocyanin synthesis and laying a foundation for the molecular breeding of tree peony.

In tree peony, the glycosides kaempferol, luteolin, apigenin, and isosalipurposide (ISP) were the main flavonoids investigated. ISP is a kind of chalcone that is the main pigment component of *P. delavayi* var. *lutea* petals [2,3]. A few genes have been identified to be involved in the formation of ISP, including chalcone synthase (*CHS*) and chalcone2′-glucosyltransferases (*THC2′GT*). In ISP biosynthesis, *CHS* catalyzes 1-molecule coumaroyl-CoA and 3-molecule malonyl-CoA to form 2′,4′,6′,4-tetrahyroxychalcone (THC), and THC generates a stable ISP in the vacuole under the action of *THC2′GT* [4]. THC produces naringenin (colorless) under the action of chalcone isomerase (*CHI*), which is the precursor of anthocyanin, copigment flavone, and flavonol biosynthesis [5,6,7]. *THC2’GT* converted THC into chalcononaringenin 2’-*O*-glucoside, which is yellow anthocyanin ISP. Because ISP is only found in a few species, such as carnation [8], cyclamen, catharanthus [9], kangaroo paw [10], and peony [2,3], *THC2′GT* has rarely been studied. Itoh et al. found that the synthesis and accumulation of *THC2’GT* were the reasons why carnation (*Dianthus caryophyllus*) petals appear yellow [4]. Yoshida et al. speculated that the difference in expression level or enzyme activity of *THC2’GT* was the main reason for the difference in ISP content (concentration ranging from 5.5% to 100%) in petals of different varieties of carnation and speculated that ISP might not be regulated by a single gene but by multiple genes [8]. Togami et al. cloned the cDNA of *THC2’GT* from carnation, *Cyclamen persicum,* and *Catharanthus roseus* and expressed it in *Petunia hybrida*, causing the accumulation of ISP in *P. hybrida* petals [11].

There is no detailed report of the *THC2’GT* gene in tree peony. We examined the effects of the candidate gene (*PdTHC2’GT*) of the *THC2’GT* gene (found in the *P. delavayi* transcriptome database (SRA No: PRJNA772706)), as well as its upstream gene *CHS* (*PdCHS*) and competitive pathway gene *CHI* (*PdCHI*). We used subcellular localization, transgenic technology, qRT-PCR, and LC-MS/MS to verify their functions. The determination of the role of *THC2’GT* is a critical step in the directional breeding of yellow tree peony varieties.

## 2. Results

### 2.1. Cloning and Sequence Analysis of PdTHC2′GT, PdCHS, and PdCHI

The full-length CDS sequences of *PdTHC2’GT*, *PdCHS*, and *PdCHI* genes were cloned by full-length primers. *PdTHC2’GT* sequences with 1428bp, *PdCHS* sequences with 1185bp, and *PdCHI* sequences with 666 bp were obtained (Figure 1a). After sequencing, *PdTHC2’GT* had a complete ORF from the start codon ATG to the stop codon TAA, encoding 475 amino acids. *PdTHC2’GT* was logged into NCBI with the accession number: OM478635. *PdCHS* and *PdCHI* had a complete ORF from the start codon ATG to the stop codon TGA, encoding 394 and 221 amino acids, respectively. *PdCHS* and *PdCHI* were logged into NCBI with the accession numbers: OM478633 and OM478632, respectively. In this study, the *2’GT*, *3GT*, *5GT*, and *7GT* of the GT family and *PdTHC2’GT* were selected to construct the phylogenetic tree (Figure 1b); phylogenetic analysis showed that *PdTHC2’GT* and *CrTHC2’GT* (*Catharanthus roseus*) and *CpTHC2’GT* (*Cyclamen persicum*) and *DcTHC2’GT* (*Dianthus caryophyllus*) were clustered in the same branch. Multiple sequence alignment analysis showed that *PdTHC2’GT* had high homology with them and was highly conserved in the conserved domain PSPG-box of glucosyltransferase (Figure 2a). In summary, *PdTHC2’GT* is a strong candidate for key structural genes involved in the synthesis of petal pigment ISP in *P. delavayi* var. *lutea*. In addition, we constructed the phylogenetic tree of *PdCHS* and *PdCHI*. The results showed that both *PdCHS* and amino acids with *CHS* function in PKS type III had high homology, which further suggested that they had chalcone synthase function (Figure 1c); *PdCHS* had a highly conserved domain of *CHS* (Figure 2b). The phylogenetic tree results showed that *PdCHI*, *Camellia fraterna,* and *Actinidia rufa* had the closest genetic relationship (Figure 1d), clustered into one branch, and they had the conserved active sites of *CHI* (Figure 2c).

The basic physical and chemical properties of the proteins encoded by *PdTHC2’GT*, *PdCHS,* and *PdCHI* were analyzed by online tools. Their protein molecular weights (PMW) were 52.25, 43.28, and 24.68 kDa, respectively. Their protein gravies (PG) were 0.014, −0.115, and −0.122, respectively. Their protein isoelectric points (Pl) were pH 6.63, pH 6.60, and pH 4.35, respectively.

The template library of SWISS-MODEL was searched by BLAST and HHBlits. It was found that PdTHC2’GT had the highest homology with the triterpenoid triterpene UDP-glucosyl transferase UGT71G1 protein model of *Medicago truncatula* in the database (44.42%), PdCHS had the highest homology with the *Oryza sativa* chalcone synthase 1 protein model in the database (82.78%), and PdCHI had the highest homology with *Arabidopsis thaliana* chalcone-flavonone isomerase protein model in the database (65.20%). Therefore, based on the above models, the 3D models of *PdTHC2’GT*, *PdCHS,* and *PdCHI* proteins were constructed (Figure 3).

### 2.2. The Expression Pattern of PdTHC2′GT, PdCHS, and PdCHI

We combined transcriptome data, real-time fluorescence quantitative PCR results, and flower color phenotype analysis in four stages of flower development. The petals were green in the S1 stage. In the S2 stage, yellow began to appear, and the petals showed yellowish green. The petals completely turned yellow during the S3 stage. It remained yellow in the S4 period, but it became lighter than that in the S3 stage (Figure 4a). The expression of *PdTHC2’GT* was upregulated in S1–S2, reached the highest in S2, and then downregulated. Due to the delayed effect of the gene expression to phenotype, we thought that the differential expression of *PdTHC2’GT* was consistent with the change in flower color. *PdCHS* was highly expressed in S1–S3, reached the highest expression in S3, and decreased slightly in S4. Because *CHS* is the first enzyme in the metabolic pathway of flavonoids, it regulates the formation of THC, the substrate of all flavonoids. The sustained high expression of *PdCHS* after flower color yellowing was in accordance with the change in flower color. The expression of *PdCHI* decreased continuously in S1–S4 and almost did not express in S4. We speculated that the expression of *PdTHC2’GT* competed with that of *PdCHI*, which made the expression of *PdCHI* lower (Figure 4b). We speculated that this was the reason why the petals of *P. delavayi* var. *lutea* did not show red.

### 2.3. Subcellular Localization of PdTHC2′GT, PdCHS, and PdCHI

Subcellular localization of *PdTHC2’GT*, *PdCHS,* and *PdCHI* was carried out using fluorescent reporter genes (GFP). The 35S::PdTHC2′GT-GFP, 35S::PdCHS-GFP, and 35S::PdCHI-GFP recombinant vector was constructed and introduced into tobacco leaf epidermal cells using 35S::GFP as the negative control, and GFP fluorescence was observed with a laser confocal microscope (ZEISS LSM880 Airyscan FAST+NLO, Carl Zeiss AG, Oberkochen, Germany). Based on the observed colocalization with the marker proteins, 35S::PsGSTF3-GFP fluorescence was present in both nucleus and cytoplasm. The fluorescence of *PdTHC2’GT*, *PdCHS,* and *PdCHI* was obviously expressed in the nucleus and cytoplasm. Thus, we speculated that these three genes are located in the nucleus and cytoplasm and may play a role in those areas (Figure 5).

### 2.4. Overexpression of PdTHC2′GT, PdCHS, and PdCHI in Tobacco

To further characterize the function of *PdTHC2′GT*, *PdCHS,* and *PdCHI*, the 35S::PdTHC2′GT-pCAMBIA1302, 35S::PdCHS-pCAMBIA1302, and 35S::PdCHI-pCAMBIA1302 vectors were ectopically introduced into tobacco. Six independent transgenic tobacco lines (line1-line6) expressing *PdTHC2’GT*, *PdCHS,* and *PdCHI* were screened in a hygromycinB-resistant medium and were then cultured under the same conditions. The DNA of each tobacco line (line1–line6) and wildtype tobacco (WT) was extracted, and the extracted DNA was used as a template to clone *PdTHC2’GT*, *PdCHS,* and *PdCHI* using full-length primers. The results of agarose gel electrophoresis showed that the transformed *PdTHC2’GT*, *PdCHS,* and *PdCHI* could be detected in their transgenic lines, but no bands were detected in wildtype tobacco (Figure 6a). The results of qRT-PCR on 18 tobacco lines showed that *PdTHC2’GT*, *PdCHS,* and *PdCHI* were all expressed in different tissues, and the three genes were the highest in stems. The expression of *PdTHC2′GT* and *PdCHI* in leaves and petals had little difference, and the expression of *PdCHS* in leaves was slightly higher than that in petals (Figure 6b).

The phenotypes of tobacco after flowering were observed. After flowering, the red color of six tobacco lines with overexpressed *PdTHC2′GT* was lighter than that of WT tobacco, and the petals were not uniformly lighter, among which the petals of line5 and line6 appeared yellow. After the flowering of the tobacco with overexpressed *PdCHS*, the color of the six tobacco lines was slightly darker than that of WT tobacco. After flowering, the red color of the six tobacco lines overexpressed with *PdCHI* was darker than that of WT tobacco and deeper than that of the tobacco lines with overexpressed *PdCHS* (Figure 7a). qRT-PCR was used to analyze the changes in *NtCHS* and *NtCHI* in tobacco petals with overexpressed *PdTHC2’GT*, *PdCHS,* and *PdCHI*. The results showed that after overexpression of *PdTHC2′GT*, the expression of *NtCHS* gen in tobacco petals was obviously upregulated (Figure 7b), and the expression of *NtCHI* was obviously downregulated (Figure 7c). The expression of *NtCHI* in tobacco petals was significantly upregulated after overexpression of *PdCHS* (Figure 7d). The expression of *NtCHS* in tobacco petals was significantly upregulated after overexpression of *PdCHI* (Figure 7e). However, since there is no THC2’GT gene in tobacco, the effect of overexpression of CHS and CHI on the THC2’GT gene was not clear. This indicated that the overexpression of *PdTHC2’GT*, *PdCHS,* and *PdCHI* in tobacco did affect the expression of genes in the tobacco flavonoid synthesis pathway.

Qualitative and quantitative analyses of flavonoids in the transgenic tobacco lines and WT tobacco overexpressed with *PdTHC2′GT* were carried out. The qualitative and quantitative analyses of anthocyanins in transgenic tobacco lines and WT tobacco overexpressed with *PdCHS* and *PdCHI* were also carried out. We further verified the consistency between the change in phenotype, the change in structural gene expression of the flavonoid synthesis pathway, and the change in pigment level of the flavonoid synthesis pathway. After absolute quantification of the anthocyanin content, the results showed that, compared with WT tobacco, most anthocyanin components in the overexpressed *PdCHS* and *PdCHI* transgenic tobacco lines were upregulated to different degrees, and the upregulation degree of anthocyanin components in *PdCHI* transgenic tobacco lines was higher than that in *PdCHS* transgenic tobacco lines, among which cyanidin, malvidin, and pelargonidin showed a significant upregulation difference, followed by delphinidin, but there was no significant difference in peonidin, petunidin, and flavonoid (Figure 7f). The extremely significant increase in the contents of cyanidin-3-*O*-rutinoside, cyanidin-3-*O*-glucoside, pelargonidin-3-*O*-rutinoside, and delphinidin-3-*O*-rutinoside-5-*O*-glucoside led to the difference in flower color (Appendix A). After the relative quantification of flavonoids, the results showed that compared with the WT tobacco, the contents of chalcone and some flavones showed a significant upward trend, and part of the anthocyanins and proanthocyanidins showed a significant downward trend. Flavanones, flavanonols, part of the anthocyanins, and flavones showed a downward trend to varying degrees. It is worth emphasizing that the yellow pigment ISP of *P. delavayi* var. *lutea* petals was detected in the transgenic tobacco petals after overexpression of *PdTHC2’GT*, which was not found in the WT tobacco petals (Appendix A). The above results proved that *PdTHC2’GT* is indeed the key structural gene for the synthesis of ISP (Figure 7g).

## 3. Discussion

Flower color is the most important ornamental character of tree peony, among which there are few yellow peony varieties, which has always been an important breeding goal of breeders. The main source of yellow pigment in tree peony petals is ISP. The structural and regulatory genes of anthocyanin biosynthesis have been widely studied in ornamental plants and tree peonies, but there are few studies on the biosynthesis of ISP in plants, especially in peonies. Therefore, it is of far-reaching significance to study the multi-molecular regulation mechanism of ISP biosynthesis in tree peony; it is an unavoidable step in the process of directional breeding of yellow tree peony varieties.

ISP is a class of flavonoids modified by a glycosyl group and located in vacuoles [12]. Its glycosylation is completed by 2’GT, a member of the glycosyltransferase (GTs) family. This gene has only been reported in a few plants, such as carnation, cyclamen, and catharanthus [4,11]. Therefore, the 2’GT is not annotated in the KEGG database. 2’GT belongs to 1 of the 109 glycosyltransferase (GTs) subfamilies. Family1 is the largest family of glycosyltransferases, which is characterized by a highly conserved motif with a C-terminal consistent sequence of 44 amino acids, which is called the plant secondary product glycosyltransferase box (PSPG-box) [12]. In addition to 2’GT, GT families in the flavonoid biosynthesis pathway include flavonoid 3-glucosyltransferase (3GT), flavonoid 5-glucosyltransferase (5GT), flavonoid 7-glucosyltransferase (7GT), and flavonoid 3’-glucosyltransferase (3’GT), 3GT and 5GT form independent branches, and 7GT and 3’GT belong to the same branch [11]. In this study, through the annotation of transcriptome data, we screened out the differentially expressed genes annotated to the GT1 family and then screened out the genes consistent with the changing trend of flower color phenotype, from which we obtained a candidate gene *PdTHC2’GT* encoding *THC2’GT*. In this study, the gene sequence encoding *THC2’GT* was cloned from the petals of *P. delavayi* var. *lutea*. Sequence analysis showed that *PdTHC2’GT* had the typical structure of glycosyltransferase PSPG-box, and cluster analysis with 3GT, 5GT, and 7GT in the flavonoid biosynthesis pathway showed that *PdTHC2’GT* and *THC2’GTs* of catharanthus, cyclamen, and carnation were clustered in the same branch. Therefore, it was preliminarily concluded that the *PdTHC2’GT* gene also had the function of synthesizing ISP.

*CHS* is the first structural gene in the flavonoid metabolic pathway and the upstream gene of *THC2′GT,* which encodes the synthesis of THC. The *CHS* gene was first found in parsley in 1983 [13] and cloned successively in *Antirrhinum majus* [14], *Arabidopsis thaliana* [15], *Sorghum bicolor* [16], and other plants. In the past studies, white plants were obtained by silencing *CHS* in *Torenia hybrida* [17], *Brunfelsia acuminata* [18], *Dahlia variabilis* [19], and *Gentiana scabra* [20]. By overexpressing *CHS* in tobacco, tobacco petals changed from pink to red [21]. Gu demonstrated that PsMYB12 interacts with a bHLH and a WD40 protein in a regulatory complex that directly activates PsCHS expression, which is also specific to the petal blotches [22]. Therefore, we speculated that the *CHS* gene of *P. delavayi* var. *lutea* can also play an important role in regulating downstream flavonoids. *CHI* is one of the most important enzymes in anthocyanin metabolism. *CHI* competes with *THC2’GT* for the THC, serving different flavonoid products. *CHI* has been cloned in many plants, and its gene function has also been verified. Inhibiting the expression of *CHI* will change the color of petals from red to white [23]. When plants share anthocyanin and ISP, the loss of *CHI* activity will lead to the increase in ISP accumulation [24]. Therefore, through transcriptome annotation, we obtained candidate genes *PdCHS* and *PdCHI* encoding CHS and CHI of *P. delavayi* var. *lutea*, and we cloned their full-length sequences for functional study.

Subcellular localization analysis is very important to study the function of proteins. Only when the location of proteins is accurate can their biological functions be exercised normally. In this study, the precise subcellular localization of *PdTHC2’GT*, *PdCHS,* and *PdCHI* was verified by fusing the C-terminal of each protein coding region with sGFP, under the control of the CaMV35S strong promoter, and transient expression in tobacco leaves. The reason for this verification was that the subcellular location prediction website algorithm is usually inaccurate [25]. When using the Cell-Ploc2.0 online prediction website (http://www.csbio.sjtu.edu.cn/bioinf/Cell-PLoc-2/, accessed on 9 July 2020) to predict the three genes, the website predicted that the *PdTHC2’GT* was located on the cell membrane, the *PdCHS* was located on the cytoplasm, and the *PdCHI* was located on the chloroplast, which was not consistent with the results of this study. In previous studies, both *CHS* and *CHI* were located in the cytoplasm and nucleus [26,27], which was consistent with the results of the subcellular localization of *PdCHS* and *PdCHI* in this study. In this study, *PdTHC2’GT* was located in the cytoplasm and nucleus. Due to the lack of previous studies on *THC2’GT*, it was impossible to find the location of *THC2’GT* in previous studies. We looked at the subcellular localization studies of other members of the GT family, and most of the other GT members were located in the cytoplasm (endoplasmic reticulum) and nucleus [28]. Therefore, we think that the Cell-Ploc2.0 subcellular localization prediction software itself has some limitations. Several other studies have reported the nuclear localization of some flavonoid bisosynthesis enzymes, which responds to in situ synthesis of flavonoids there [26,29,30]. The biosynthesis of flavonoids in the nucleus may serve to protect DNA from UV and oxidative damage [31]. In summary, *PdTHC2’GT, PdCHS,* and *PdCHI* were distributed in the cytoplasm and nucleus, which further indicates that the biosynthesis of flavonoids in petals of *P. delavayi* var. *lutea* was complex. We speculate that there are many flavonoid biosynthesis sites in *P. delavayi* var. *lutea*, and further research is needed to understand the regulation mechanism in different location sites.

In order to clarify the role of *PdTHC2’GT*, *PdCHS,* and *PdCHI* in the biosynthesis of flavonoids from *P. delavayi* var. *lutea*, *PdTHC2’GT*, *PdCHS,* and *PdCHI* were transformed into tobacco. We detected the flower, stem, and leaf of transgenic tobacco lines by qRT-PCR. Surprisingly, the expression of the transferred gene was the highest in the stem, but the stem color of transgenic tobacco lines was not different from that of WT tobacco. We speculated that the main pigment in the stem was chlorophyll, and the accumulation of flavonoid substrates and products were not enough to cause a color change. The low expression level of flowers may be related to the sampling stage. In this study, the sampling stage of petals by qRT-PCR was the initial opening period of tobacco petals, and the tobacco petals had already produced red pigment. Because of the delayed effect of genes, it was not the highest point of gene expression at this time. In this experiment, the effects of the *PdTHC2’GT*, *PdCHS,* and *PdCHI* genes on the expression of *CHS* and *CHI* in the tobacco flavonoid pathway were studied. Overexpression of *PdTHC2’GT* in tobacco can increase the expression of *NtCHS* and decrease the expression of *NtCHI*. This showed that *THC2’GT* is obviously competitive with *CHI*, which is consistent with Forkmann and Dangelmayr’s research results [24].

After overexpression of *PdCHS* and *PdCHI* in tobacco, the petal color of tobacco changed from pink to dark pink. LC-MS/MS data and phenotype showed that overexpressed *PdCHI* produced more anthocyanins than overexpressed *PdCHS*. We speculated that *PdCHI* contributed more to anthocyanin formation. After overexpression of *PdTHC2’GT* in tobacco, the color of the tobacco petals became lighter, the transgenic line1-line4 did not produce obvious yellow, line6 petals produced slight yellow, and line5 produced obvious yellow. LC-MS/MS data showed that ISP was produced in tobacco petals after overexpression of *PdTHC2’G*, which did not exist in WT tobacco, which proved that *PdTHC2’GT* indeed regulated the synthesis of ISP. Togami cloned *THC2’GT* from carnation, cyclamen, and *Catharanthus roseus* and then overexpressed them in *Petunia hybrida*. The results showed that these genes caused the accumulation of ISP in *Petunia hybrida* petals but did not cause the petals of *Petunia hybrida* to turn yellow [11]. This is similar to the results of this study, and we speculate that it is because the accumulation of ISP has not reached the chromogenic level. Yoshida et al. speculated that the difference in *THC2’GT* expression or enzyme activity was the main reason for the difference in ISP content in the petals of different varieties of carnation [8]. Therefore, we speculate that increasing the expression or enzyme activity of *PdTHC2’GT* can increase the content of isoberellin in petals, resulting in high accumulation and yellow petals. Inhibiting the expression of *CHI*, improving the expression efficiency of *PdTHC2’GT*, and efficiently accumulating ISP to make petals appear yellow are the main research directions for the future.

## 4. Materials and Methods

### 4.1. Plant Materials

*P. delavayi* var. *lutea* is an important breeding resource for tree peonies with yellow flowers. All the materials were collected from flowers of tree peony in Shangri-La County, Yunnan, China (E: 99°34′60″; N: 27°58′7″). The development of the yellow flowers was distinguished into four stages from buds to open flowers: unpigmented tight bud (S1), slightly pigmented soft bud (S2), initially opened flower (S3), and fully opened flower (S4) (Figure 4a). Only petals were collected as experimental materials; they were immediately frozen in liquid nitrogen and then stored at −80 °C.

### 4.2. Gene Cloning and Sequence Analysis

By searching the sequence annotation information of the petal full-length transcriptome database constructed earlier, the suspected sequences of *PdTHC2′GT*, *PdCHS,* and *PdCHI* were obtained, and the CDS regions of the three genes were obtained by using the NCBI open reading frame tool ORFfinder (https://www.ncbi.nlm.nih.gov/orffinder/, accessed on 2 March 2020). Primer3web online primer design software was used for primer design (https://primer3.ut.ee/, accessed on 2 March 2020). PrimeSTAR^®^ HS DNA Polymerase (TaKaRa, Beingjing, China) was used for gene cloning. The primers and restriction sites are listed in Appendix A.

Bioinformatics analysis software and tools were used to predict the physical and chemical properties, structure, and function of proteins encoded by *PdTHC2′GT*, *PdCHS,* and *PdCHI*, so as to provide reference for further research and application of *PdTHC2′GT*, *PdCHS,* and *PdCHI*. The basic physicochemical properties of proteins encoded by *PdTHC2′GT*, *PdCHS,* and *PdCHI* were analyzed by the Prot-Param tool in ExPasy (https://web.expasy.org/protparam/, accessed on 17 March 2020). The conserved domains of proteins encoded by *PdTHC2′GT*, *PdCHS,* and *PdCHI* were predicted by the CD-Search function on NCBI (https://www.ncbi.nlm.nih.gov/Structure/cdd/wrpsb.cgi, accessed on 17 March 2020). The online tool Swiss-model (https://swissmodel.expasy.org/, accessed on 17 March 2020) was used to predict the three-dimensional structure of P *PdTHC2′GT*, *PdCHS,* and *PdCHI* proteins. The amino acid sequence of *PdTHC2’GT**, PdCHS,* and *PdCHI* were analyzed by DNAMAN software 7.0 and the BLAST tool in NCBI. MEGA7.0 software was used to construct the phylogenetic tree of *PdTHC2’GT**, PdCHS,* and *PdCHI* by Neighbor-Joining. The protein sequences used to construct phylogenetic trees are shown in Appendix A.

### 4.3. Gene Expression Analysis

The quantitative real-time polymerase chain reaction (qRT-PCR) was performed using the SYBR Premix Ex TaqTM II Kit (TaKaRa, China) on a LightCycler 480 system (Roche Applied Science, Penzberg, Germany). *PdTHC2′GT*, *PdCHS,* and *PdCHI* were analyzed by qRT-PCR in the S1-S4 stages, when the petals of *P. delavayi* var. *lutea* opened. After stable transformation in tobacco, the expression levels of *PdTHC2′GT*, *PdCHS,* and *PdCHI* in tobacco flowers, leaves, and stems were analyzed by qRT-PCR. The expression of *NtCHS* and *NtCHI* in transgenic tobacco lines was analyzed by qRT-PCR. The 2^−ΔΔCT^ method was used for analysis and visualization of the qRT-PCR data generated by multiple technical replicates. The *PP2A* gene was used as a reference gene for tree peony. The *NtCP* gene was used as a reference gene for tobacco. The primers and restriction sites are listed in Appendix A. 

### 4.4. Subcellular Localization Analyses

The coding sequence (CDS) of *PdTHC2′GT*, *PdCHS,* and *PdCHI* without stop codons were cloned into PHG vectors (modified from PHB vector) to construct the 35S::PdTHC2′GT-GFP, 35S::PdCHS-GFP, and 35S::PdCHI-GFP recombinant vectors using a seamless cloning kit (Novoprotein, Suzhou, China). Then, the fusion constructs and negative control 35S::GFP vector were transformed into Agrobacterium strain GV3101. Additionally, the empty-mCherry [32] was used to locate the fluorescent proteins in the cytoplasm and nucleus, and 35S::PdTHC2′GT-GFP or 35S::PdCHS-GFP or 35S::PdCHI-GFP or 35S::GFP vector (1:1, *v*/*v*) were infiltrated into *N. benthamiana* leaves. The fluorescence signal was observed by laser confocal microscopy (ZEISS LSM880 Airyscan FAST+NLO, Carl Zeiss AG, Oberkochen, Germany). The primers and restriction sites are listed in Appendix A.

### 4.5. Tobacco Stable Transformation

The coding sequence (CDS) of *PdTHC2′GT*, *PdCHS,* and *PdCHI* with stop codons were cloned into pCAMBIA1302 vectors to construct the 35S::PdTHC2′GT-pCAMBIA1302, 35S::PdCHS-pCAMBIA1302, and 35S::PdCHI-pCAMBIA1302 recombinant vectors using a seamless cloning kit. Then, the fusion constructs and negative control pCAMBIA1302 vector were transformed into Agrobacterium strain EHA105. We took the leaves of healthy tobacco tissue culture seedlings, trimmed off the veins and edges, cut them into 0.5 × 0.5 mm size, soaked some of the cut leaves in the infection solution, soaked some of them in the liquid culture for 1/2 MS as a blank control, and shook them continuously for 5 min to make the leaves fully contact with the bacterial solution. After infection, the leaves consecutively carried out differentiation culture and rooting culture on an MS medium supplemented with the relevant hormones (different concentrations of IBA and NAA) and antibiotics (hygromycinB). Finally, the well-growing plantlets were transplanted into small flowerpots containing the substrate to maintain the original growth environment. Six tobacco seedlings from different lines were randomly selected, and their positive seedlings were identified. Tobacco DNA was extracted by using the DNA extraction kit (Tiangen, China), and then the tobacco seedlings were tested positively. The PCR primers are shown in Appendix A, and the results of the agarose gel electrophoresis with corresponding gene bands were the positive seedlings. For the specific experimental methods, refer to Horsch et al. [33].

### 4.6. Extraction and Detection of Anthocyanins

The sample was freeze-dried, ground into powder (30 Hz, 1.5 min), and stored at −80 °C until needed. Then, 50 mg powder was weighed and extracted with 0.5 mL methanol/water/hydrochloric acid (500:500:1, *v*/*v*/*v*). Next, the extract was vortexed for 5 min, underwent ultrasound for 5 min, and centrifuged at 12,000× *g* at 4 °C for 3 min. The residue was re-extracted by repeating the above steps under the same conditions. The supernatants were collected and filtered through a membrane filter (0.22 μm, Anpel) before LC-MS/MS analysis. 

The sample extracts were analyzed using an UPLC-ESI-MS/MS system (UPLC, ExionLC™ AD, Sciex, Shanghai, China; MS, Applied Biosystems 6500 Triple Quadrupole, Sciex, Shanghai, China). The analytical conditions were as follows: UPLC: column, WatersACQUITY BEH C18 (1.7 µm, 2.1 × 100 mm); solvent system, water (0.1% formic acid): methanol (0.1% formic acid); gradient program, 95:5 *v/v* at 0min, 50:50 *v/v* at 6 min, 5:95 *v/v* at 12 min, hold for 2 min, 95:5 *v/v* at 14 min; hold for 2min; flow rate, 0.35 mL/min; temperature, 40 °C; injection volume, 2 μL.

Anthocyanins contents were detected by MetWare (http://www.metware.cn/, accessed on 8 February 2022). For the specific experimental methods, refer to Romina et al. [34,35].

### 4.7. Extraction and Detection of Flavonoids

The biological samples were freeze dried using a vacuum freeze-dryer (Scientz-100F, Scientz, Ningbo, China). The freeze-dried sample was crushed using a mixer mill (MM 400, Retsch, Arzberg, Germany) with a zirconia bead for 1.5 min at 30 Hz. We dissolved 100 mg of lyophilized powder with 1.2 mL 70% methanol solution, vortexed for 30 s every 30 min, 6 times in total, and placed the sample in a refrigerator at 4 °C overnight. Following centrifugation at 12,000 rpm for 10 min, the extracts were filtered (SCAA-104, 0.22 μm pore size; ANPEL, Shanghai, China) before UPLC-MS/MS analysis.

The sample extracts were analyzed using an UPLC-ESI-MS/MS system (UPLC, SHIMADZU Nexera X2, Shimadzu, Kyoto, Japan; MS, Applied Biosystems 4500 Q TRAP, Thermo Fisher Scientific, Waltham, MA, USA). The analytical conditions were as follows, UPLC: column, Agilent SB-C18 (1.8 µm, 2.1 × 100 mm); the mobile phase consisted of solvent A, pure water with 0.1% formic acid, and solvent B, acetonitrile with 0.1% formic acid. Sample measurements were performed with a gradient program that employed the starting conditions of 95% A, 5% B. Within 9 min, a linear gradient to 5% A, 95% B was programmed, and a composition of 5% A, 95% B was kept for 1 min. Subsequently, a composition of 95% A, 5.0% B was adjusted within 1.1 min and kept for 2.9 min. The flow velocity was set as 0.35 mL per minute; the column oven was set to 40 °C; and the injection volume was 4 μL. The effluent was alternatively connected to an ESI-triple quadrupole-linear ion trap (QTRAP)-MS.

Flavonoid contents were detected using MetWare (http://www.metware.cn/, accessed on 8 February 2022). For specific experimental methods, refer to Chen et al. [36,37,38].

## 5. Conclusions

Sequence analysis and functional verification of *PdTHC2′GT*, *PdCHS,* and *PdCHI* showed that *PdCHS* and *PdCHI* regulated the accumulation of anthocyanin in flavonoid biosynthesis, and *PdTHC2′GT* was found to be the key structural gene for synthesizing ISP. The overexpression of *PdTHC2′GT* caused the accumulation of ISP in tobacco petals. The qRT-PCR analysis of *NtCHS* and *NtCHI* in transgenic tobacco lines showed that *THC2’GT* increased the expression of *CHS*, and *THC2’GT* was found to be in competition with *CHI*; hence, the overexpression of *THC2’GT* could decrease the expression of *CHI*. *CHS* and *CHI* can increase each other’s expression.

## Figures and Tables

**Figure 1 ijms-23-05696-f001:**
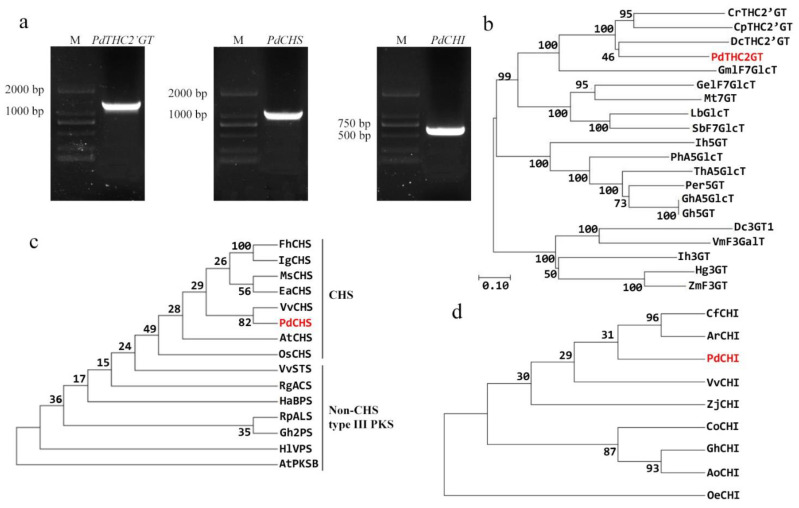
Sequence and phylogenetic analysis. (**a**) Polymerase chain reaction (PCR) amplification products of *PdTHC2′GT*, *PdCHS,* and *PdCHI*. M is DNA marker DL2000. (**b**) Phylogenetic tree analysis of glycosyltransferases. (**c**) Phylogenetic tree analysis of chalcone synthase. (**d**) Phylogenetic tree analysis of chalcone isomerase. The protein sequences are listed in Appendix A. Red font is the gene of *P. delavayi* var. *lutea*.

**Figure 2 ijms-23-05696-f002:**
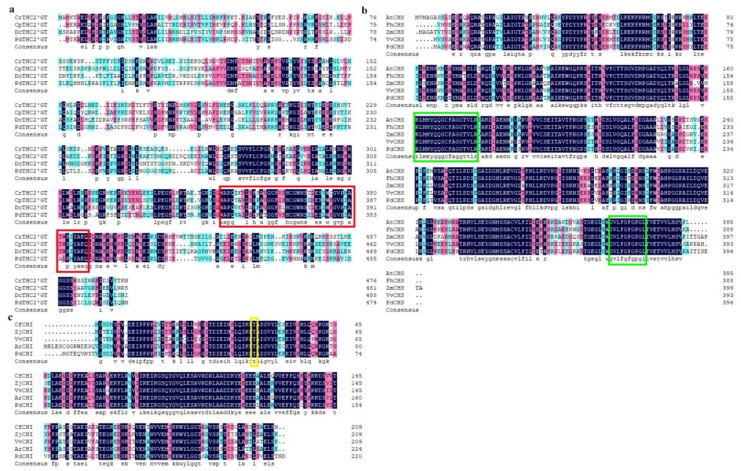
Sequence alignment of *THC2′GT*, *CHS*, and *CHI* from *P. delavayi* var. *lutea* and other species. (**a**) Amino acid sequences’ homologous analysis of the *THC2′GT* gene and other species’ related genes. The red box shows the glycosyltransferase domains—PSPG-box. (**b**) Amino acid sequences’ homologous analysis of the *CHS* gene and other species’ related genes. The green box indicates the highly conserved domains of *CHS*. (**c**) Amino acid sequences’ homologous analysis of the *CHI* gene and other species’ related genes. The yellow box indicates the conserved active sites of *CHI*.

**Figure 3 ijms-23-05696-f003:**
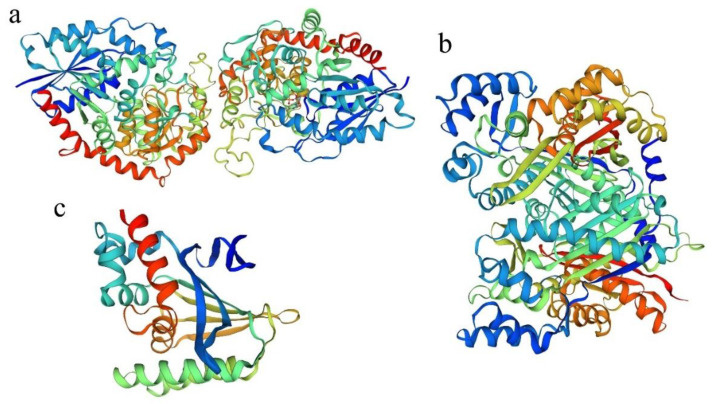
The protein modeling results of PdTHC2′GT, PdCHS, and PdCHI. (**a**) Visualizable 3D model of PdTHC2′GT. (**b**) Visualizable 3D model of PdCHS. (**c**) Visualizable 3D model of PdCHI.

**Figure 4 ijms-23-05696-f004:**
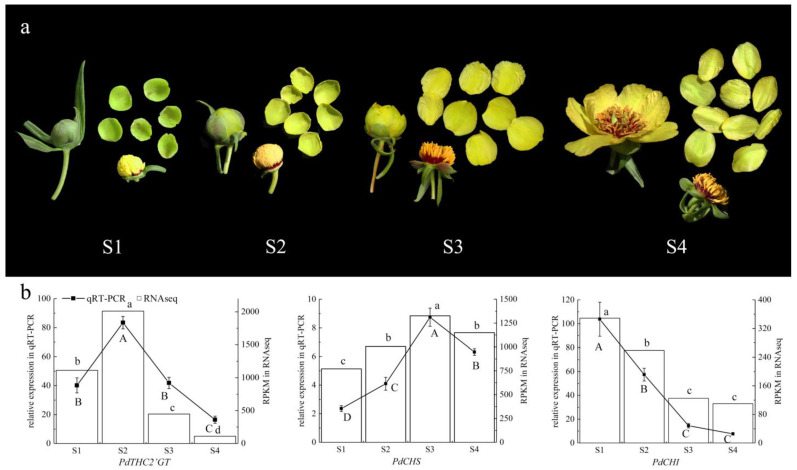
The expression patterns of *PdTHC2′GT, PdCHS,* and *PdCHI*. (**a**) Four blooming periods of *P. delavayi* var. *lutea* flowers. S1: Unpigmented tight bud, the petals are completely green. S2: Slightly pigmented soft bud, the petals begin to change from green to yellow. S3: Initially opened flower, the stamens are not visible, and the petals have completely turned yellow. S4: Fully opened flower, the stamens are completely exposed, and the petals are a lighter yellow than those in S3. (**b**) Comparison of RNA-seq and qRT–PCR analyses for *PdTHC2′GT*, *PdCHS,* and *PdCHI*. The different lowercase letters indicate significant differences in RNAseq among the different sampling points, the different capital letters indicate significant differences in qRT-PCR among the different sampling points (*p* < 0.05), and the error bars indicate the standard deviations.

**Figure 5 ijms-23-05696-f005:**
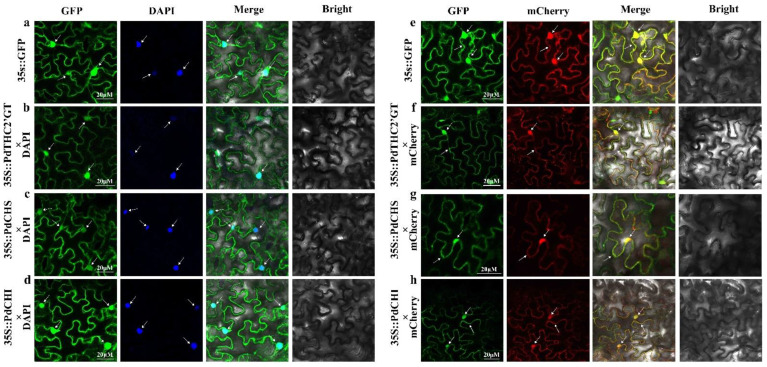
Subcellular localization of *PdTHC2′GT*, *PdCHS,* and *PdCHI* in *Nicotiana benthamiana*. (**a**) Subcellular localization in *N. benthamiana* infected with 35S::GFP and DAPI. (**b**) *N. benthamiana* infected with 35S::PdTHC2′GT-GFP and DAPI. (**c**) *N. benthamiana* infected with 35S::PdCHS-GFP and DAPI. (**d**) *N. benthamiana* infected with 35S::PdCHI-GFP and DAPI. (**e**) *N. benthamiana* infected with 35S::GFP and mCherry. (**f**) *N. benthamiana* infected with 35S::PdTHC2′GT-GFP and mCherry. (**g**) *N. benthamiana* infected with 35S::PdCHS-GFP and mCherry. (**h**) *N. benthamiana* infected with 35S::PdCHI-GFP and mCherry. DAPI was used as a nucleus marker, mCherry was used as a cytoplasm and nucleus marker.

**Figure 6 ijms-23-05696-f006:**
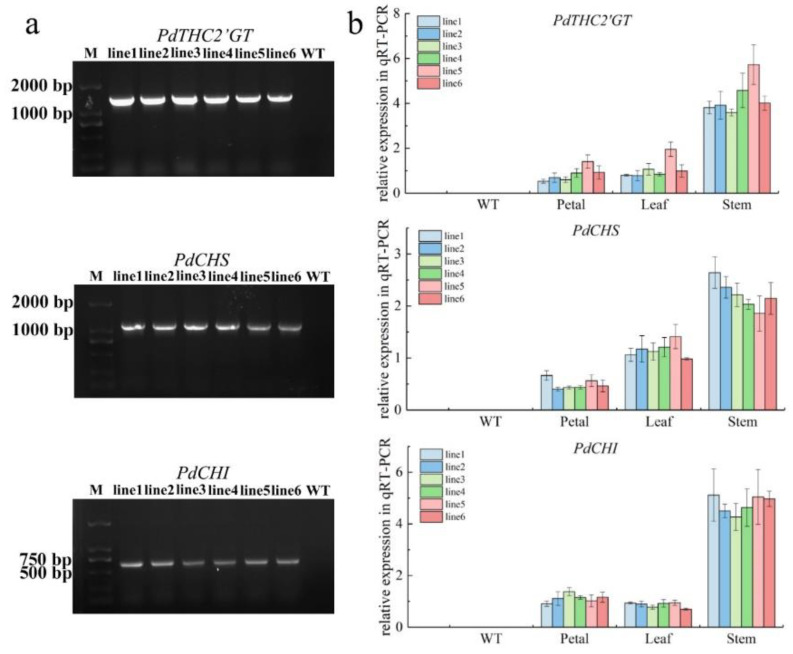
Positive detection of transgenic tobacco. (**a**) Electropherogram of positive PCR detection in transgenic tobacco lines. (**b**) Temporal and spatial expression patterns of *PdTHC2’GT*, *PdCHS,* and *PdCHI* in the six transgenic tobacco lines. Data are the means (±SD) from three biological replicates.

**Figure 7 ijms-23-05696-f007:**
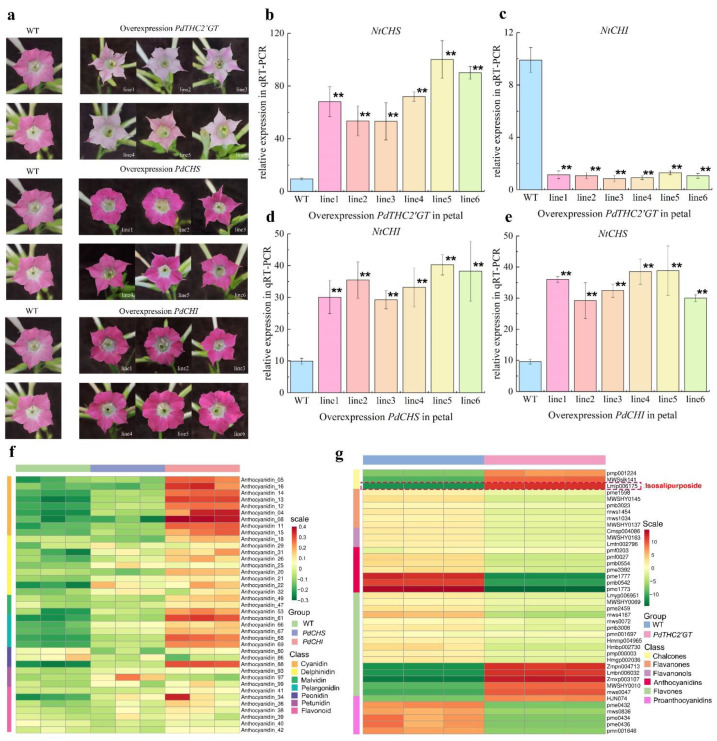
Functional analysis of tobacco lines overexpressing *PdTHC2’GT*, *PdCHS,* and *PdCHI*. (**a**) Phenotypic characterization of tobacco flowers between the wildtype group and transgenic lines. (**b**) After overexpression of *PdTHC2’GT*, the expression quantity of *NtCHS* in tobacco flowers changed. (**c**) After overexpression of *PdTHC2’GT*, the expression quantity of *NtCHI* in tobacco flowers changed. (**d**) After overexpression of *PdCHS*, the expression quantity of *NtCHI* in tobacco flowers changed. (**e**) After overexpression of *PdCHI*, the expression quantity of *NtCHS* in tobacco flowers changed. (**f**) Heatmap analysis of anthocyanin component. (**g**) Heatmap analysis of flavonoids component. The contents of (**f**,**g**) were log-transformed and used to generate a heatmap with the TBtools software package (v0.66443, Ziping Yang, Guangdong, China, open resources). The asterisks denote significant differences according to a one−way analysis of variance (ANOVA) (** *p* < 0.01).

## Data Availability

The transcriptome data link is https://www.ncbi.nlm.nih.gov/sra/PRJNA772706 (PRJNA772706), accessed on 27 October 2022.

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
