# Peer review of "Sequence Analysis and Functional Verification of the Effects of Three Key Structural Genes, PdTHC2’GT, PdCHS and PdCHI, on the Isosalipurposide Synthesis Pathway in Paeonia delavayi var. lutea"

_ijms, 2022, doi:10.3390/ijms23105696_

Round 1
Reviewer 1 Report
This manuscript does provide strong and sound evidence for the gene that they cloned (PdTHC2'GT ) to be responsible for the yellow pigmentation in peony flowers. There are not a lot of questions that I would have. However, writing needs lots of improvement. There are many verbose, lengthy and repetitive sentences in lots of places, need to tidy up and try to break long sentence into short and clear sentences. Also, line number should be added in so that the reviewer can indicate the places where edition needs to be made. I have made my comments in the attached pdf as well.
Few questions as the following:
“ Login PdCHS and PdCHI to NCBI with the accession number: OM478633, OM478632” This is not a full sentence, rewrite.
Why do not do the the phylogenetic analysis for PdCHI and PdCHS genes? You cloned and sequenced three genes, provide expression and localisation data, but only do the phylogenetic type of analysis for PdTHC2'GT?
Questions: So overexpressing CHS leads to upregulating of CHI, but not THC2'GT in tobacco? This part is not clear to me. It seems CHS is upstream of both CHI and THC2'GT, so why overexpress CHS only up-regulate CHI and not THC2'GT? There should be a homologous THC2'GT in tobacco or it does not because it does not have yellow flower color? You still needs to blast the gene and state the results (i.e. no homologous gene for THC2'GT was found for tobacco). If there is, then I would like to see the expression data for this gene.
This is strange that the expression of the upstream gene CHS is in tune with the phenotype (protein expression) while the pDTHC2'GT was off (earlier, it was lowest when the protein of THC is supposed to be highest to give the brightest colour in the flowers). There is strong evidence that CHS negatively and directly regulates CHI. But I don't know why the correlation between pDTHC2'GT and phenotype (flower colour) is very weak although the protein level of ISP was up-regulated.

Author Response
Reviewer 1
This manuscript does provide strong and sound evidence for the gene that they cloned (PdTHC2'GT ) to be responsible for the yellow pigmentation in peony flowers. There are not a lot of questions that I would have. However, writing needs lots of improvement. There are many verbose, lengthy and repetitive sentences in lots of places, need to tidy up and try to break long sentence into short and clear sentences. Also, line number should be added in so that the reviewer can indicate the places where edition needs to be made. I have made my comments in the attached pdf as well.
Dear Reviewer:
Thanks for your letter and for the comments concerning our manuscript entitled “Sequence Analysis and functional Verification of three key structural genes PdTHC2'GT, PdCHS, PdCHI on Isosalipurposide Synthesis Pathway in Paeonia delavayi var. lutea” (Manuscript Number: ijms-1699475). Those comments are all valuable and very helpful for revising and improving our paper, as well as the important guiding significance to our researches. We have studied comments carefully and have made correction which we hope meet with approval. Revised portion are marked in red in the manuscript (revised version).
About language, we seek the help from a professional company (MDPI) to further improve the language of this whole paper and revised seriously according to the requirements, and finally gained a language editing certificate. But the language modification was not reflected in red in the manuscript.
The main corrections in the paper and the responds to the reviewer’s comments are as follows. And it’s noting that the black and red font represents the comment and response in this paper, respectively.
Major:
- Login PdCHS and PdCHI to NCBI with the accession number: OM478633, OM478632” This is not a full sentence, rewrite.
Response: Thanks for your suggestion. I have corrected the sentence to: PdCHS and PdCHI was logged into NCBI with the accession number: OM478633, OM478632.
2.Why do not do the the phylogenetic analysis for PdCHI and PdCHS genes? You cloned and sequenced three genes, provide expression and localisation data, but only do the phylogenetic type of analysis for PdTHC2'GT?
Response: CHS and CHI are widely studied in plants, and there are few subtypes in the family, so these two genes can be directly annotated by transcriptome data, which is the reason why CHS and CHI have not been phylogenetic analyzed and sequence aligned in the original text. Following your advice and adding this part can make the structure of the article more complete. I modified Figure 1, added Figure 2, and added the description of this part, as follows:
Figure 1. Sequence and phylogenetic analysis. (a) Polymerase chain reaction (PCR) amplification products of PdTHC2’GT, PdCHS, and PdCHI. M is DNA marker DL2000. (b) Phylogenetic tree analysis of glycosyltransferases. (c) Phylogenetic tree analysis of chalcone synthase. (d) Phylogenetic tree analysis of chalcone isomerase. The protein sequences are listed in Table S2.
Figure 2. Sequence alignment of THC2’GT, CHS, and CHI from P. delavayi var. lutea and other species. (a) Amino acid sequences’ homologous analysis of the THC2’GT gene and other species related genes. The red box shows the glycosyltransferase domains—PSPG-box. (b) Amino acid sequences’ homologous analysis of the CHS gene and other species’ related genes. The green box indicates the highly conserved domains of CHS. (c) Amino acid sequences’ homologous analysis of the CHI gene and other species’ related genes. The yellow box indicates the conserved active sites of CHI.
In this study, the 2'GT, 3GT, 5GT, and 7GT of the GT family and PdTHC2'GT were selected to construct the phylogenetic tree (Figure 1b); phylogenetic analysis showed that PdTHC2'GT and CrTHC2'GT (Catharanthus roseus) and CpTHC2'GT (Cyclamen persicum) and DcTHC2'GT (Dianthus caryophyllus) were clustered in the same branch. Multiple sequence alignment analysis showed that PdTHC2'GT had high homology with them and was highly conserved in the conserved domain PSPG-box of glucosyltransferase (Figure 2a). In summary, PdTHC2'GT is a strong candidate for key structural genes involved in the synthesis of petal pigment ISP in P. delavayi var. lutea. In addition, we constructed the phylogenetic tree of PdCHS and PdCHI. The results showed that both PdCHS and amino acids with CHS function in PKS type III had high homology, which further suggested that they had chalcone synthase function (Figure 1c); PdCHS had a highly conserved domain of CHS (Figure 2b). The phylogenetic tree results showed that PdCHI, Camellia fraterna, and Actinidia rufa had the closest genetic relationship (Figure 1d), clustered into one branch, and they had the conserved active sites of CHI (Figure 2c).
- So overexpressing CHS leads to upregulating of CHI, but not THC2'GT in tobacco? This part is not clear to me. It seems CHS is upstream of both CHI and THC2'GT, so why overexpress CHS only up-regulate CHI and not THC2'GT? There should be a homologous THC2'GT in tobacco or it does not because it does not have yellow flower color? You still needs to blast the gene and state the results (i.e. no homologous gene for THC2'GT was found for tobacco). If there is, then I would like to see the expression data for this gene.
Response: There is no THC2'GT in tobacco, which has not been searched in gene database or reported in related literature. This can also be proved when the pigment of wild-type tobacco petals is measured. There was no ISP in tobacco petals. Therefore, overexpression of CHS in tobacco will lead to up-regulation of the expression of anthocyanin pathway gene CHI in tobacco itself. I have added a note to the article, as follows:
qRT-PCR was used to analyze the changes in NtCHS and NtCHI in tobacco petals with overexpressed PdTHC2'GT, PdCHS, and PdCHI. The results showed that after overexpression of PdTHC2′GT, the expression of NtCHS gen in tobacco petals was obviously upregulated (Figure 7b), and the expression of NtCHI was obviously downregulated(Figure 7c). The expression of NtCHI in tobacco petals was significantly upregulated after overexpression of PdCHS (Figure 7d). The expression of NtCHS in tobacco petals was significantly upregulated after overexpression of PdCHI (Figure 7e). However, since there is no THC2'GT gene in tobacco, the effect of overexpression of CHS and CHI on the THC2'GT gene was not clear. This indicated that the overexpression of PdTHC2'GT, PdCHS, and PdCHI in tobacco did affect the expression of genes in the tobacco flavonoid synthesis pathway.
4.This is strange that the expression of the upstream gene CHS is in tune with the phenotype (protein expression) while the pDTHC2'GT was off (earlier, it was lowest when the protein of THC is supposed to be highest to give the brightest colour in the flowers). There is strong evidence that CHS negatively and directly regulates CHI. But I don't know why the correlation between pDTHC2'GT and phenotype (flower colour) is very weak although the protein level of ISP was up-regulated.
Response: We thought that it is due to the delayed effect of gene expression to phenotype, and the expression of THC2'GT reached the maximum in the early stage of yellow production. CHS is the first enzyme in flavonoid metabolism pathway, and its synthesized substrate will be converted into kaempferol and quercetin besides ISP. It is reasonable that its expression level is always high. The pigment of tobacco petals over-expressed with PdTHC2'GT was detected, and the results showed that ISP was produced, which proved the function of PdTHC2'GT. The results also showed that compared with other flavonoids in tobacco, the accumulated amount of ISP was low, which was not enough to produce bright yellow. This may be related to heterologous transformation, or it may be because this pathway did not exist in tobacco originally, and all transformation efficiency is low. In the later study, we may be able to increase the accumulation of PdTHC2'GT by increasing the expression activity of ISP to achieve the purpose of color rendering.
- However, writing needs lots of improvement. There are many verbose, lengthy and repetitive sentences in lots of places, need to tidy up and try to break long sentence into short and clear sentences.
Response: Thanks for your suggestion. Based on the suggestions of reviewers and the recommendations of IJMS, we seek the help from a professional company (MDPI) to further improve the language of this paper. Then, we revised the whole paper seriously according to the requirements, and gained a language editing certificate.

Reviewer 2 Report
The study by Zou et al examined three key genes PdTHC2'GT, PdCHS, PdCHI on isosalipurposide synthesis pathway in paeonia delavayi var. lutea using various sequence analysis and functional studies. Overall, the paper is well structured and presented with novel data to support their conclusions. I have some minor suggestions.
1. To validate the genetic function of PdTHC2'GT, PdCHS, PdCHI on isosalipurposide synthesis pathway, knockout or knockdown approach should be used. This would be the most direct functional study.
2. Figure 1b, c. The figures are hard to read. High-resolution figures are needed.
3. Figure 2. 1) In the main context, the authors simply stated that PdTHC2'GT, PdCHS and PdCHI have highest homology with respective protein models. How the 3D structural models are constructed should be clearly explained.
2) As the authors showed the model, it would be informative to include catalytic site and substrate binding mode in their structures.
3) There are mistakes that symbols for genes are italicized whereas symbols for proteins are not italicized.
4. The authors speculated that the expression of PdTHC2'GT competed with that of PdCHI. What is the molecular mechanisms regulating the differential expression patterns?
Author Response
Reviewer 2
The study by Zou et al examined three key genes PdTHC2'GT, PdCHS, PdCHI on isosalipurposide synthesis pathway in paeonia delavayi var. lutea using various sequence analysis and functional studies. Overall, the paper is well structured and presented with novel data to support their conclusions. I have some minor suggestions.
Dear Reviewer:
Thanks for your letter and for the comments concerning our manuscript entitled “Sequence Analysis and functional Verification of three key structural genes PdTHC2'GT, PdCHS, PdCHI on Isosalipurposide Synthesis Pathway in Paeonia delavayi var. lutea” (Manuscript Number: ijms-1699475). Those comments are all valuable and very helpful for revising and improving our paper, as well as the important guiding significance to our researches. We have studied comments carefully and have made correction which we hope meet with approval. Revised portion are marked in red in the manuscript (revised version).
The main corrections in the paper and the responds to the reviewer’s comments are as follows. And it’s noting that the black and red font represents the comment and response in this paper, respectively.
Major:
- To validate the genetic function of PdTHC2'GT, PdCHS, PdCHI on isosalipurposide synthesis pathway, knockout or knockdown approach should be used. This would be the most direct functional study.
Response: Thanks for your suggestion. Since there is no genetic transformation system in tree peony, the experimental method of gene knockout is difficult to realize. later, we will consider using VIGS technology to carry out instantaneous silencing of Paeonia delavayi var. lutea petals to verify the function of genes.
- Figure 1b, c. The figures are hard to read. High-resolution figures are needed.
Response: Thanks for your suggestion. We have modified the picture and enlarged the font to make it clear. Now the picture is as follows:
Figure 1. Sequence and phylogenetic analysis. (a) Polymerase chain reaction (PCR) amplification products of PdTHC2’GT, PdCHS, and PdCHI. M is DNA marker DL2000. (b) Phylogenetic tree analysis of glycosyltransferases. (c) Phylogenetic tree analysis of chalcone synthase. (d) Phylogenetic tree analysis of chalcone isomerase. The protein sequences are listed in Table S2.
Figure 2. Sequence alignment of THC2’GT, CHS, and CHI from P. delavayi var. lutea and other species. (a) Amino acid sequences’ homologous analysis of the THC2’GT gene and other species related genes. The red box shows the glycosyltransferase domains—PSPG-box. (b) Amino acid sequences’ homologous analysis of the CHS gene and other species’ related genes. The green box indicates the highly conserved domains of CHS. (c) Amino acid sequences’ homologous analysis of the CHI gene and other species’ related genes. The yellow box indicates the conserved active sites of CHI.
- Figure 2. 1) In the main context, the authors simply stated that PdTHC2'GT, PdCHS and PdCHI have highest homology with respective protein models. How the 3D structural models are constructed should be clearly explained.
Response: With regard to the construction of 3D structural model, SWISS-MODEL is a very convenient website for model construction. Only by submitting amino acid sequences, the optimal matching and construction can be carried out automatically. The SEQ ID is over 30%, and the result of model construction is credible, so we don't use complicated methods to build the model, and we don't give a detailed explanation in this article.
2) As the authors showed the model, it would be informative to include catalytic site and substrate binding mode in their structures.
Response: Thanks for your suggestion. To be honest, I am not familiar with the field of protein modeling. After querying the literature, adding catalytic site and substrate binding mode may require molecular docking. I'm not sure if I can master this technique in a short time, and I'm not sure if I can get the right catalytic site and substrate binding mode. I implore the experts whether it is possible not to add this part without having a great impact on the logic and results of the article.
3) There are mistakes that symbols for genes are italicized whereas symbols for proteins are not italicized.
Response: I have checked and revised the full text of the article, and all descriptions of protein have been changed from italics to orthography.
- The authors speculated that the expression of PdTHC2'GT competed with that of PdCHI. What is the molecular mechanisms regulating the differential expression patterns?
Response: THC2'GT and CHI catalyze the common substrate THC to produce different products (ISP and anthocyanin), and the decrease of upstream substrate will affect the expression of downstream genes. THC2'GT conversion of THC to ISP will lead to the decrease of THC flow to anthocyanin synthesis, thus reducing the expression of CHI. This result has also been verified in carnation (References 24). As for whether THC2'GT and CHI directly inhibit the expression of each other, it has not been reported, which needs to be verified by follow-up experiments such as yeast two-hybrid.
